# Avenanthramide C Prevents Neuronal Apoptosis via PI3K/Akt/GSK3β Signaling Pathway Following Middle Cerebral Artery Occlusion

**DOI:** 10.3390/brainsci10110878

**Published:** 2020-11-20

**Authors:** Baoyuan Jin, Hyehyun Kim, Jeong-Il Choi, Hong-Beom Bae, Seongtae Jeong

**Affiliations:** 1Department of Anesthesiology and Pain Medicine, Chonnam National University Medical School and Hospital, 42 Jebong-ro, Gwangju 61469, Korea; jby631053862@gmail.com (B.J.); khh2493@gmail.com (H.K.); jichoi@jnu.ac.kr (J.-I.C.); 2The Brain Korea 21 Project, Center for Biomedical Human Resources, Chonnam National University, Gwangju 61469, Korea

**Keywords:** avenanthramide-c, cerebral ischemia, apoptosis, blood brain barrier integrity, phosphoinositide 3-kinase, Akt

## Abstract

Avenanthramides are a group of phenolic alkaloids that have been shown to have anti-inflammatory, anti-oxidant, anti-atherogenic, and vasodilation effects. The aim of the present study was to investigate the neuroprotective effect of avenanthramide-c (Avn-c) in focal brain ischemia and reperfusion injury using middle cerebral artery occlusion (MCAo) model with mice. Male C57BL/6 mice were divided into 4 groups: sham, control (MCAo), Avn-c, and Avn-c + LY294002 (phosphoinositide 3-kinase inhibitor) group. They were subjected to 60 min MCAo followed by reperfusion. Brain infarct volume and neurological deficit scores were measured after 24 h of reperfusion. We evaluated the blood brain barrier (BBB) integrity (ZO-1, VE-cadherin and occludin) and apoptosis (Bax, Bcl2, caspase3, Cytochrome C, and poly ADP ribose polymerase(PARP)-1). We also measured GSK3β for evaluation of the downstream mechanism of Akt. We examined the effect of the Avn-c in the phosphoinositide 3-kinase pathway. Avn-c reduced neurological score and infarction size. Avn-c inhibited the MCAo-induced disruption of tight junction proteins. Avn-c decreased apoptotic protein expression (Bax, Cytochrome C, and cleaved PARP-1) and increased anti-apoptotic protein expression (Bcl2) after MCAo. Akt and GSK3β were decreased in MCAo group and were restored in Avn-c group. This effect of Avn-c was abolished by PI3K inhibitor. In summary, Avn-c showed neuroprotective effects through PI3K-Akt-GSK3β signaling pathway.

## 1. Introduction

Stroke is the third leading cause of death worldwide, and the most common cause of disability [1]. Stroke results in alterations in a number of complex mechanisms, including metabolism, apoptosis, and blood–brain barrier function [2,3,4]. Endothelial tight junction proteins, such as Zonula occludens-1 (ZO-1) and occludin, as well as endothelial adherent proteins, such as vascular endothelial (VE)-cadherin, are important for maintaining the integrity of the blood–brain barrier [5]. Recent studies have shown that apoptosis plays an important role in the pathophysiologic changes caused by ischemia. Apoptosis mediates tissue preservation, neurological outcomes, and long-term survival [6,7]. Ischemia can cause an imbalance of the Bax/Bcl2 expression. Bax (pro-apoptotic regulator) and Bcl2 (anti-apoptotic regulator) play an important role in apoptosis; Bax upregulation has been observed following ischemia [8] in association with reduced expression of Bcl2. Increased Bcl2 expression is thought to protect against neuron loss after focal ischemia [9]. The release of cytochrome c from mitochondria into the cytosol triggers apoptosis, consistent with Bax translocation, followed by caspase-3 activation [10].

The neuroprotective role of phosphatidylinositol-3-kinase (PI3K)/Akt (protein kinase B, PKB) has been widely studied [11,12]. PI3Ks are a family of proteins involved in intracellular signal transduction and the regulation of cell survival [13]. Akt, a serine protein kinase, which acts downstream of PI3K, regulates cellular proliferation and metabolism process [14]. Once Akt is phosphorylated, it triggers phosphorylation of several enzymes (e.g., glycogen synthase kinase 3 beta (GSK3β), IκB kinase (IKK), the mammalian target of rapamycin (mTOR)), thereby regulating a series of cellular functions [15,16]. Previous studies showed PI3K/Akt pathway plays a critical role to regulate cell activities, inflammatory response and apoptosis in cerebral ischemia [17,18]. Glycogen synthase kinase 3 (GSK3) participates genetic expression, cellular fate, and survival. Phosphorylation of serine 9 residue in GSK3β inhibits GSK3 kinase activity [19,20]. GSK3 is considered to be linked to the pathophysiology of various neurological and neurodegenerative diseases [21,22].

Avenanthramides (Avns) are unique low molecular weight soluble phenolic compounds extracted from oats [23]. More than 25 Avn compounds have been identified. Avenanthramide c(Avn-c), which is one of the three major Avns, consists of the amide conjugates of anthranilic acid (or its hydroxylated derivatives) and hydroxycinnamic acid [24,25,26]. Avns have a strong capacity to inhibit the adhesion molecules of vascular endothelial cells, resulting in anti-inflammatory and anti-oxidative effects. A recent study showed that intake of grains contributes to a decreased risk of myocardial infarction [27]. However, the effects of Avns on brain ischemia and the underlying mechanism thereof are unknown. Herein, we examined the effects of Avn-c, including anti-apoptosis effects, in a mouse model of middle cerebral occlusion.

## 2. Materials and Methods

### 2.1. Animal Preparation and Study Model

#### 2.1.1. Animals and Study Approval

Wild-type (C57BL/6J, 8 weeks old) mice were purchased from Samtako (Osan, Korea). In all experiments, male mice were used. The mice were housed in individually ventilated cages with free access to food and water. The breeding room was controlled under a 12 h light/12 h dark cycle, with the light phase beginning at 8:00 p.m. The temperature was maintained at 22–30 °C. Animals were acclimatized to the laboratory environment for more than 1 week prior to injury. All experiments involving animals followed protocols approved by the Institutional Animal Care and Use Committee of Chonnam National University Medical School, Republic of Korea (CNU IACUC-H-2018-36).

#### 2.1.2. Middle Cerebral Artery Occlusion (MCAo) Model

The MCAo model was generated as described previously [28]. After anesthesia induction under 3% sevoflurane, the trachea was intubated with a 20-gauge intravenous catheter (Becton-Dickinson, Sandy, UT, USA), and the lungs were mechanically ventilated with 1.6% sevoflurane(Hana Pharm, Seoul, Korea) in 30% O_2_ and 70% N_2_. Rectal temperature was maintained at 37 ± 0.2 °C, and the right common carotid artery, the right external carotid artery, and the internal carotid artery were exposed through a ventral midline neck incision. A 6-0 monofilament silicone rubber-coated monofilament (L56PK10; Doccol Corporation, Sharon, MA, USA) was introduced into the right common carotid artery and advanced until slight resistance was felt. Reperfusion was achieved by withdrawing the suture after 60 min of occlusion, to restore blood supply to the middle cerebral artery (MCA) territory. Body temperature was maintained at 36.5–37.5 °C throughout the procedure, i.e., from the start of surgery until the animals recovered.

#### 2.1.3. Drug Dosing and Delivery

Avn-c was obtained from Sigma-Aldrich (St. Louis, MO, USA) and dissolved in dimethylsulfoxide (DMSO) (St. Louis, MO, USA). The PI3K inhibitor LY294002 was also obtained from Sigma-Aldrich (St. Louis, MO, USA). Prior to injury, mice were randomized to 3 or 4 groups depend on experiments sham group; control group (20 μL DMSO); Avn-c group (100 mg/kg/20 μL DMSO); and LY294002 + Avn-c group (20 mM/20 μL DMSO). Avn-c was administered via intraperitoneal injection immediately after MCAo. In the LY294002 group, LY294002 was administered 1 h before MCAo, followed by administration of Avn-c. After 24 h of reperfusion, the mice were anesthetized and sacrificed.

### 2.2. Evaluation of Neurologic Function and Infarct Volume

Animal behavior was evaluated by foot-fault test at different time points based on modified Bederson scoring methods: 0, no deficit; 1, forelimb flexion; 2, forelimb flexion plus decreased resistance to lateral push; 3, unidirectional circling; 4, longitudinal spinning or seizure activity; 5, no movement [29]. Evaluation was performed by a laboratory assistant blinded to the experimental design. The average score of each experimental mouse from three trials was used for statistical analysis. Mice were anesthetized and brains were removed. Brains with subarachnoid hemorrhage and/or clot formation in the MCA were excluded from the analysis. All brains were sliced into 1-mm sections. Slices were incubated for 30 min in a 0.1% solution of 2,3,5-triphenyltetrazolium chloride (TTC; Sigma-Aldrich, St. Louis, MO, USA) at 37 °C and fixed in 10% buffered formaldehyde solution(Sigma-Aldrich, St. Louis, MO, USA). For analysis, the cross-sectional area of the infarct in the right MCA territory of each brain slice was determined with ImageJ software (Java 1.8.0_170) (NIH, MD, USA). The total mean infarct area of each section was calculated as the average of the area on the rostral and caudal surface. The hemispheric lesion volume was calculated by multiplying the area by slice thickness. The areas of the infarcted tissue, and the areas of both hemispheres, were calculated on each brain slice.

### 2.3. Western Blot Analysis

Mice were sacrificed after 1 h of ischemia followed by 24 h of reperfusion. The right hemisphere was quickly removed and pulverized into powder in liquid nitrogen. Brains were homogenized in lysis buffer. Protein concentration was determined by the Bio-Rad protein assay (Bio-Rad, Hercules, CA, USA). Lysates (30 μg) were electrophoresed using 10% SDS-PAGE (Bio-Rad, CA, USA) and transferred to nitrocellulose membranes, and Western blot analysis was carried out. Blots were developed with the ECL chemiluminescence system (GE Healthcare, Piscataway, NJ, USA) and captured on autoradiographic films (Kodak, Rochester, NY, USA). Films were scanned and densitometric analysis of the bands was performed with AlphaEase Image Analysis FC 4.0 Software (Alpha Innotech Corp., San Leandro, CA, USA).

### 2.4. Statistical Analysis

All data are means ± standard error of means (SEM) of at least three independent experiments. For statistical analysis, n indicates the number of independent experiments. All statistical analyses were performed by one-way analysis of variance, followed by Tukey’s multiple comparison post-hoc test using SPSS 23.0 software (IBM, CHI, USA). A *p* value of <0.05 was considered statistically significant.

## 3. Results

### 3.1. Avn-c Treatment Improved Neurologic Function and Reduced Cerebral Infarct Volume and Inhibition of PI3K Signaling Abolished the Effects of Avn-c on Cerebral Infarction

Avn-c treatment significantly decreased neurological deficit score. TTC staining showed that mice treated with Avn-c exhibited reduced infarct volume compared with the control group at 24 h after MCAo. To determine the role of PI3K in MCAo mice, we treated mice with the PI3K inhibitor LY294002. Pretreatment with the PI3K inhibitor LY294002 abolished the neuroprotective effects of Avn-c (Figure 1).

### 3.2. Avn-c Prevents MCAo Induced Disruption of the Blood-Brain Barrier

ZO-1, VE-cadherin, and occludin are located in tight junctions, whose major role to mediate endothelial cells contact integrity [30,31]. Structural integrity of endothelial cells is crucially important to the blood–brain barrier. Disruption of the blood–brain barrier is observed in strokes and causes alteration of endothelial cells integrity [32]. Western blot analysis of protein samples from the ischemic hemisphere showed that brain damage was significantly reduced in Avn-c-treated MCAo mice via protection of the blood–brain barrier (Figure 2).

### 3.3. Avn-c Prevents MCAo Induced Inhibition of the PI3K/Akt Pathway

PI3K/Akt signaling plays a role in ischemic stroke; this signaling pathway is frequently disrupted in stroke patients. To examine the possible mechanism of Avn-c in MCAo mice, we determined the expression of phosphorylated (p)-PI3K, p-Akt, and p-GSK3β. Expression levels of p-PI3K, p-Akt, and p-GSK3β were significantly decreased in MCAo compared with Sham group. Avn-c significantly increased p-PI3K, p-Akt, and p-GSK3β compared with MCAo group. These data suggest that MCAo-induced brain damage attenuated the expression of PI3K/Akt/GSK3β and Avn-c significantly reversed the MCAo-induced changes in protein expression. The phosphorylation levels were reversed following treatment with LY294002 (PI3K inhibitor) (Figure 3). These data indicated that PI3K signaling played an important role in our mouse model of ischemia, and that Avn-c reversed focal ischemia in a PI3K-dependent manner.

### 3.4. Avn-c Attenuated MCAo-Induced Changes in Apoptotic and Anti-Apoptotic Proteins and Anti-Apoptosis Markers Were Reduced Following Treatment with a PI3K Inhibitor

Inhibition of apoptosis is an important defensive strategy for preventing brain damage in the event of ischemia. The expression of Bax/Bcl2, cytochrome c, cleaved caspase-3, and cleaved PARP-1 in ischemic hemisphere tissue were analyzed by Western blotting 24 h after MCAo. MCAo significantly increased apoptotic protein (Bax, cytochrome c, cleaved caspase-3) and cell death (cleaved PARP-1), and decreased anti-apoptotic protein (Bcl2). In the Avn-c group, there were significant downregulation of Bax and cytochrome c, and significant upregulation of Bcl2, observed as compared with MCAo group (control). Similarly, the protein levels of cleaved caspase-3 and cleaved PARP1 were markedly increased in MCAo mice compared with the sham group, but were reduced in MCAo mice treated with Avn-c. To determine whether LY294002 altered the anti-apoptotic effects of Avn-c, we pretreated mice with LY294002 1 h prior to MCAo. The Bax/Bcl2 ratio and protein levels of cytochrome c were significantly suppressed by Avn-c; however, treatment with LY290042 abolished the effects of Avn-c. As expected, phosphorylation of caspase-3 and PARP-1 was increased in LY2094002-treated mice compared with Avn-c-treated mice (Figure 4).

## 4. Discussion

In stroke cases, limiting neurological damage and improving neuronal recovery requires rapid intervention; intervention within 3 h of a stroke is critical for prognosis. In this study, treatment with Avn-c decreased both blood–brain barrier disruption and the infarct size 24 h after MCAo. We observed a neuroprotective effect of Avn-c via activation of the PI3K/Akt/GSK3β pathway, as well as anti-apoptotic effects. PI3K was significantly phosphorylated at 24 h following administration of Avn-c in the MCAo model. However, pre-administration of the PI3K inhibitor LY294002 abolished the neuroprotective effects of Avn-c; LY294002 also attenuated the anti-apoptotic effects of Avn-c.

Apoptosis is a homeostatic mechanism to control the cell population. It also plays an important role following brain damage due to ischemia. Studies have shown that protein synthesis may reduce hippocampus neuron death, where the majority of such damage occurs within 3 days of focal ischemia [33,34]. Apoptosis plays an important role in brain ischemia injury, by increasing the final size of the infarction [35,36]. In the present study, apoptosis was exacerbated in MCAo mice due to the disruption of mitochondrial function via upregulation of Bax, cytochrome c, and cleaved caspase-3. Avn-c treatment attenuated upregulation of these proteins and activated a cell survival protein, Bcl2, thereby preventing apoptosis. When mice were treated with LY294002, protein levels of apoptotic markers were reversed, consistent with MCAo mice; the anti-apoptotic effects of Avn-c were almost abolished. These data suggest that Avn-c may influence the Bcl2 family to protect the ischemic brain from apoptosis.

The PI3K pathway has been widely studied; clinical trials of PI3K inhibitors as putative cancer treatments are ongoing [37]. PI3K, and its related intracellular signaling pathway, is important in the immune response and cell regulation [38]. Previous studies have reported that the PI3K/Akt pathway participates in apoptosis and might be a therapeutic target for brain ischemia, by reducing the final size of the infarction. GSK3β is a multifaceted protein involved in neuronal development, metabolism, and cell survival [39,40]. Phosphorylation at Ser9, as mediated by Akt, inactivates GSK3β. Inactive GSK3β has been shown to prevent apoptosis, and abnormal phosphorylation of GSK3 is thought to cause neurodegenerative diseases [41,42]. In the present study, phosphorylation of PI3K/Akt/GSK3β was attenuated in MCAo mice; however, administration of Avn-c significantly upregulated protein levels of PI3K/Akt/GSK3β. Treatment with the PI3K inhibitor LY294002 suppressed phosphorylation of Akt and GSK3β. Thus, PI3K plays an important role in protecting the brain from MCAo.

In summary, our study demonstrated that Avn-c showed the neuroprotective effect and inhibited the ischemia-induced disruption of tight junction proteins in MCAo model. This neuroprotective effect of Avn-c may be through the PI3K/Akt/GSK3β signaling pathway related to reducing apoptotic protein.

## Figures and Tables

**Figure 1 brainsci-10-00878-f001:**
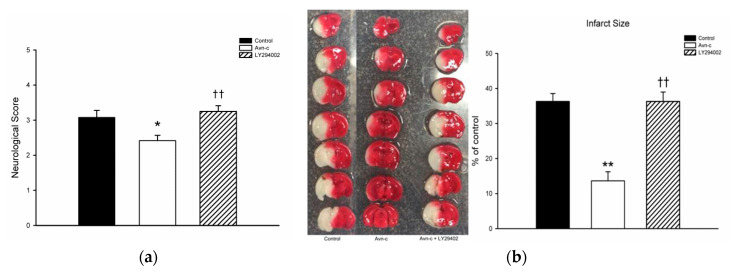
Avn-c treatment improved neurologic function and reduced cerebral infarct volume l: (**a**) Avn-c treatment significantly decreased neurological deficit scores (data from 36 mice, *n* = 12); (**b**) Representative photographs of TTC-stained brain sections (1 mm) showing the infarction area from MCAo, Avn-c, and Avn-c + LY294002 (PI3K inhibitor) groups at 24 h after MCAo (data from 12 mice, *n* = 4). Quantitative analysis of infarction size by TTC staining and presented as a percentage of the total brain volume. Data shown are the mean ± SEM. * *p* < 0.05 and ** *p* < 0.01 vs. Control group, ^††^
*p* < 0.01 vs. Avn-c group.

**Figure 2 brainsci-10-00878-f002:**
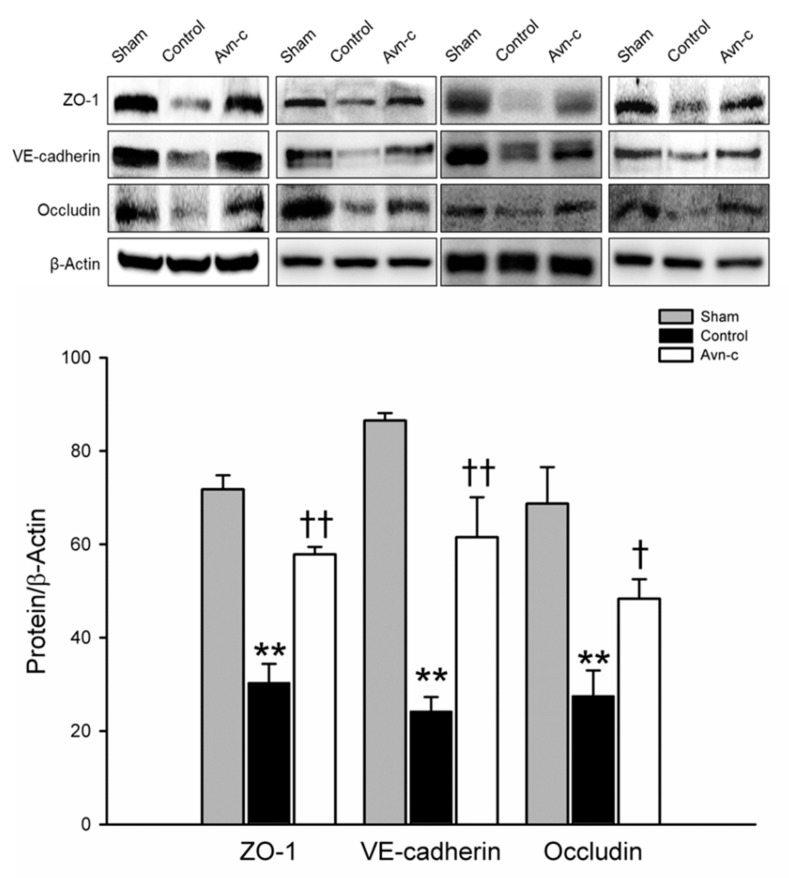
Western blot analysis for the expression of ZO-1, VE-cadherin and Occludin. The quantification of ZO-1, VE-cadherin and occludin protein expression. Data shown are mean ± SEM (data from 12 mice, *n* = 4). ** *p* < 0.01 vs. Sham group, and ^†^
*p* < 0.05, ^††^
*p* vs. Control group.

**Figure 3 brainsci-10-00878-f003:**
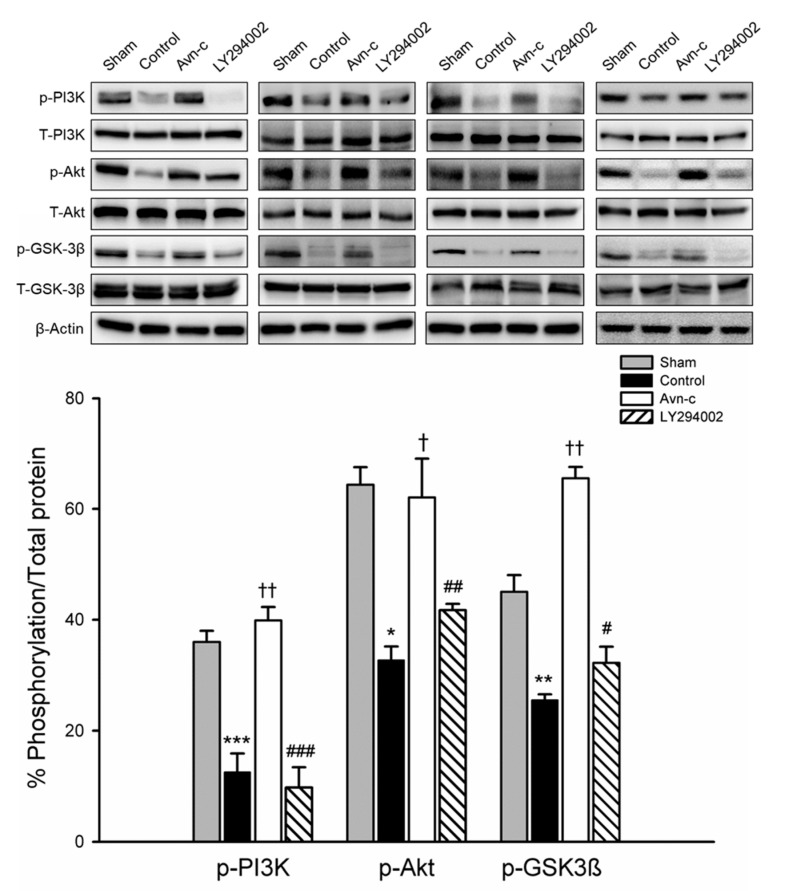
Avn-c prevents MCAo induced inhibition of the PI3K/Akt pathway. Expression of phosphorylated (p)-PI3K, p-Akt, and p-GSK3β. Quantitative analysis of protein expression changes in different groups. The quantification of ZO-1, VE-cadherin and occludin protein expression. Data shown are mean ± SEM (data from 16 mice, *n* = 4). * *p* < 0.05, ** *p* < 0.01 and *** *p* < 0.001 vs. sham group. ^†^
*p* < 0.05 and ^††^
*p* < 0.01 vs. control group. # *p* < 0.05, ## *p* < 0.01 and ### *p* < 0.001 vs. Avn-c group.

**Figure 4 brainsci-10-00878-f004:**
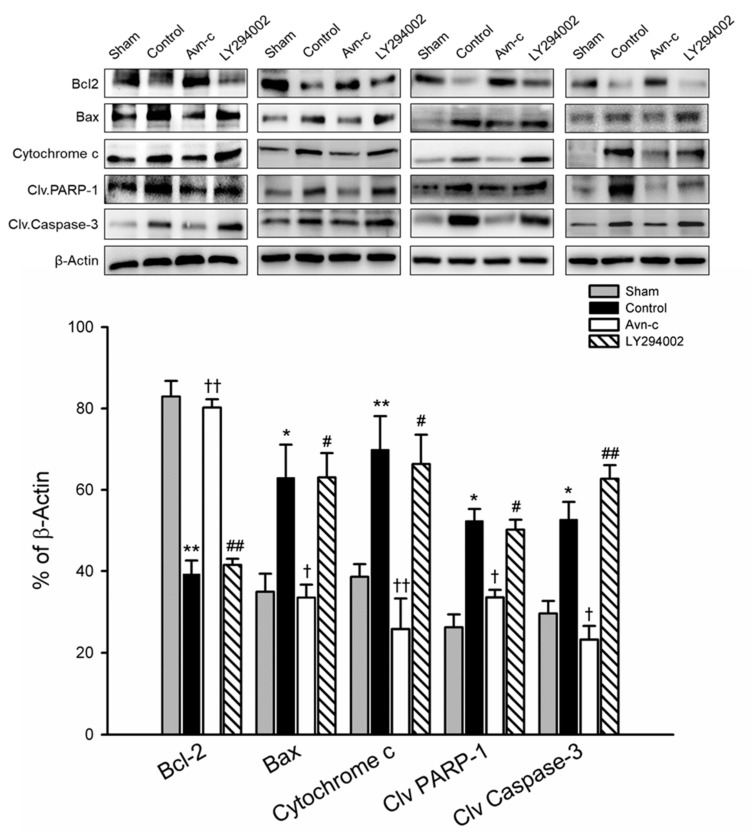
Avn-c exhibits anti-apoptotic effect in MCAo mice. Western blot analysis for the expression of Bcl-2, Bax, Cytochrome c, Cleaved PARP-1 and Cleaved caspase-3. Quantitative analysis of Bcl2, Bax, Cytochrome c, Cleaved PARP-1 and Cleaved caspase-3 expression. Data shown are mean ± SEM (data from 16 mice, *n* = 4). * *p* < 0.05 and ** *p* < 0.01 vs. Sham group. ^†^
*p* < 0.05 and ^††^
*p* < 0.01 vs. Control group. # *p* < 0.05, ## *p* < 0.01 vs. Avn-c group.

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
