# Peer review of "Avenanthramide C Prevents Neuronal Apoptosis via PI3K/Akt/GSK3β Signaling Pathway Following Middle Cerebral Artery Occlusion"

_brainsci, 2020, doi:10.3390/brainsci10110878_

Round 1
Reviewer 1 Report
In this study the authors examine the effects of Avenanthramide C using a mouse model of middle cerebral occlusion. They state that the effects and molecular mechanisms of action of Avenanthramides on brain ischemia are unknown and for this reason these results are novel and of mechanistic importance. The authors state that Avn-c reduced neurological score, infarction size and prevents MCAo induced disruption of the blood-brain barrier. However, I have concerns regarding the way the statistical analysis was done in these experiments and as such I am unsure if the results presented are statistically significant. Therefore, the authors need to provide a clear explanation regarding this aspect . It is my opinion that this manuscript needs major revisions. In addition, the manuscript needs editing of the English language.
Line 12: In the abstract the authors should explain what type of drug is Avenanthramide C or what are Avenanthramides, before mentioning their effects.
For instance: Avenanthramides are a group of phenolic alkaloids that have been shown to have anti-inflammatory, anti-oxidant, anti-atherogenic and vasodilation effects.
Line 15: Where it reads: “Male C57BL/6 mice were allocated 2 group:” it should read: Male C57BL/6 mice were divided into 2 groups:
Line 38: “Stroke is the third leading cause of death worldwide, and the most common cause of disability.”-the authors need to add reference(s).
Lines 53-54: Where it reads: “Akt, which acts downstream of PI3K, protect cells against apoptosis.”; it should read: Akt, which acts downstream of PI3K, protects cells against apoptosis.
Lines54-55: “Activation of Akt via phosphorylation by several enzymes (e.g., glycogen synthase kinase 3 beta [GSK3β])”. The authors mention several enzymes, however they only state one enzyme. The authors should include other examples of enzymes that phosphorylate and activate Akt.
Lines 56-58: “In a previous study, PI3K/Akt pathway that is involved in cerebral ischemic injury plays a critical role to regulate cell activities, inflammatory response and apoptosis {Nozaki, 2001 #22; Fruman, 2002 #21}.” This paragraph needs editing of the English language.
Lines 126-130: In the Statistical analysis section, the authors do not state how many independent experiments were analysed.
Line 172: Is Figure 1 the result of only 1 independent experiment? For statistical analysis there should be at least 3 independent experiments. It is not clear how the authors analysed the data. They should gather the results of each independent experiment. Average these results and then do the error bars and statistical analysis from the average of these results. For instance, I am assuming that the experiment was done 3 times (because they mention 12 mice per group, and in the Material and Methods they state 4 mice per group). It is not correct to use N=12 for the statistical analysis. They need to average each independent experiment using the 4 mice per group/experiment, and then use N=3 (from the 3 independent experiments) to calculate the statistics. This means that it is not correct to use N=12 to calculate the statistical value of their findings, unless they did the entire experiment 12 independent times.
I am concerned with the statistical significance in Figure 1a, since the error bars overlap, what makes me believe that there may be issues with how the statistical analysis was performed as mentioned above.
The same applies to the remaining Figures, since the authors mention: “Data shown are the mean ± SD from 5 individual mice in each group.” Which makes me believe that the experiment was only conducted once and as such statistical analysis cannot be conducted. The experiment needs to be done at least 3 independent times and the N is the number of independent experiments and not the number of animals.
Line 140. “ZO-1, VE-cadherin, and occludin are considered markers of blood-brain barrier integrity.” The authors need to add references here. They also need to clarify and explain how changes of these proteins in the ischemic hemisphere prove that this occurs due to the disruption of the blood brain barrier. In this case, references corroborating this need to be added.
Line 184. In Figure 3 the levels of P-Akt in LY294002 mice is similar to what is observed for Avn-C and sham mice, left panel. However, in the quantification graph they seem to indicate the contrary (non-matching results). This is concerning. How do the authors explain this?
Lines 206-207. The authors conclude: “In the present study, apoptosis was exacerbated in MCAo mice due to the disruption of mitochondrial function via upregulation of Bax, cytochrome c, and cleaved caspase-3.” Although they have done western blotting for the apoptotic proteins, apoptosis assay was performed. The authors need to add apoptosis assays in this study.
General comments:
The authors need to check and use the references format for the journal.
Author Response
We deeply appreciate your deliberate review of our manuscript. We have tried to supplement and improve the contents of the original manuscript according to the comments.
"Please see the attachment"

Reviewer 2 Report
The paper written by Baoyuan Jin and colleagues represents a study of neuroprotective properties of Avenanthramide C in mouse model of MCAo. Authors suggest that Avn-c neuroprotective properties might be related to inhibition of apoptosis.
Main criticism
- All conclusions are based on sole methodological approach – analyses of protein expression by WB. Thus such conclusion as “Avn-c improved BBB integrity after MCAo (line 22)” sound to be insufficiently supported by data. Either it has to be proved by another method or it has to be rephrased.
- In order to analyze quality of WB data and results reproducibility gels for WB have to be done including at least three replicate per experimental group. I.e. Fig 2 – Sham group: mouse1, mouse2, mouse3; Control group: mouse1, mouse2, mouse3; Anv-c group: mouse1, mouse2, mouse3.
Minor
Some sentences have to be checked for scientific meaning.
I.e. line 46 – “Bax/Bcl2, a pro-apoptotic regulator, plays an important role in apoptosis;”
Who is a pro-apoptotic regulator? Bax or Bcl2?
Line 55-56 - The neuroprotective role of phosphatidylinositol-3-kinase (PI3K)/Akt* has been widely studied 51 {Noshita, 2001 #23;Shibata, 2002 #24}.
- a) *phosphatidylinositol-3-kinase (PI3K)/Akt signaling pathway
- b) Please spell out Akt
Please add references:
Line 45 - Imbalance of the Bax/Bcl2 ratio can cause 45 ischemia [?].
References in the text are unformatted making difficult to find an appropriate one.
Author Response

(The authors gave the same response as above.)

Round 2
Reviewer 1 Report
In this study the authors examine the effects of Avenanthramide C using a mouse model of middle cerebral occlusion. They state that the effects and molecular mechanisms of action of Avenanthramides on brain ischemia are unknown and for this reason these results are novel and of mechanistic importance. The authors conclude that Avn-c reduced neurological score and infarction size.
Although the authors have made many improvements to the original manuscript. I still have major concerns regarding the statistical analysis of the results in this manuscript, and recommend major revisions.
In this second version of the manuscript the explanation of the calculation of the P values raises even more concerns regarding the way the statistical analysis in the manuscript was done. For this reason and in order for this manuscript to be considered for publication the statistical analysis for the whole manuscript needs to be re-done according to the following guidelines specified in this Nature manuscript:
Vaux, D. Know when your numbers are significant. Nature 492, 180–181 (2012).
I would like to highlight parts of this manuscript that are relevant to the statistical analysis:
“Understanding the rudiments of statistics would stop experimental biologists from calculating a P value and a s.e.m. from triplicates from one representative experiment, and might stop the reviewers and editors from letting these pass unquestioned. If the results from one representative experiment are shown, then N = 1 and statistics do not apply. Besides, it is always better to include a full data set, rather than withholding results that are not representative. When N is only 2 or 3, it would be more transparent to just plot the independent data points, and let the readers interpret the data for themselves, rather than showing possibly misleading P values or error bars and drawing statistical inferences.”
“experiments in which replicates were performed, only the mean of the replicates should be shown as a single independent data point. For replicates, no statistics should be shown, because they give only an indication of the fidelity with which the replicates were created: they might indicate how good the pipetting was, but they have no bearing on the hypothesis being tested.”
In summary, when calculating the P value only the independent experiments should be used as N values and not the total number of animals! The number of animals in each experiment are the replicates and as indicated above give only an indication of fidelity but have no bearing on the hypothesis being tested and as such no statistics should be shown for the replicates.
For example: The authors state: (n=12/group, 3 groups total n =36). This is incorrect, if the experiment was done three times, this is the number of independent experiments which is the N value. Meaning that N=3 and not 36!
This can have a huge impact in the actual significance of the results and needs to be corrected.
The N number, meaning N=?, should be clearly stated in each figure legend and should reflect the number of independent experiments only (and not the number of animals or other replicates within each experiment).
The statistical analysis section also needs clarification in this regard.
In order for this manuscript to be accepted for publication the statistical analysis of the whole manuscript needs to be redone and re-inserted taking into account the comments above.
Comments:
Where it reads: “:Avenanthramides are a group of phenolic alkaloids that have been shown anti inflammatory, antioxidant, anti atherogenic and vasodilation effects.” It should read:Avenanthramides are a group of phenolic alkaloids that have been shown to have anti inflammatory, antioxidant, anti atherogenic and vasodilation effects.
Where it reads: “Male C57BL/6 mice were allocate d divided into 4 groups: Sham, c ontrol (MCAo), Avn c, and Avn c +LY294002 (phosphoinositide 3 kinase inhibitor) group”. It should read: Male C57BL/6 mice were divided into 4 groups: Sham, c ontrol (MCAo), Avn c, and Avn c +LY294002 (phosphoinositide 3 kinase inhibitor) group
Author Response
We deeply appreciate your deliberate review of our manuscript. We have tried to supplement and improve the contents of the revised manuscript according to the comments.
"Please see the attachment"

Reviewer 2 Report
The manuscript has been significantly improved.
Author Response
We deeply appreciate your deliberate review of our manuscript. We have tried to supplement and improve the contents of the revised manuscript according to the comments.